# Modulation of VEGF Expression and Oxidative Stress Response by Iodine Deficiency in Irradiated Cancerous and Non-Cancerous Breast Cells

**DOI:** 10.3390/ijms21113963

**Published:** 2020-05-31

**Authors:** Jessica Vanderstraeten, Bjorn Baselet, Jasmine Buset, Naziha Ben Said, Christine de Ville de Goyet, Marie-Christine Many, Anne-Catherine Gérard, Hanane Derradji

**Affiliations:** 1Pole of Morphology, Institut de Recherche Expérimentale et Clinique (IREC), Université catholique de Louvain (UCL), 1200 Brussels, Belgium; bensaidnasiha@gmail.com (N.B.S.); christine.deville@uclouvain.be (C.d.V.d.G.); marie-christine.many@uclouvain.be (M.-C.M.); 2Radiobiology Unit, Belgian Nuclear Research Centre (SCK•CEN), 2400 Mol, Belgium; bjorn.baselet@sckcen.be (B.B.); jasmine.buset@sckcen.be (J.B.); hanane.derradji@sckcen.be (H.D.); 3Service d’Endocrino-Diabétologie, Centre Hospitalier Régional (CHR) Mons-Hainaut, 7000 Mons, Belgium; AnneCatherine.GERARD@chrmh.be

**Keywords:** radiation, iodine deficiency, breast, VEGF, ROS, oxidative stress, MCF12A, MCF7

## Abstract

Breast cancer remains a major concern and its physiopathology is influenced by iodine deficiency (ID) and radiation exposure. Since radiation and ID can separately induce oxidative stress (OS) and microvascular responses in breast, their combination could additively increase these responses. Therefore, ID was induced in MCF7 and MCF12A breast cell lines by medium change. Cells were then X-irradiated with doses of 0.05, 0.1, or 3 Gy. In MCF12A cells, both ID and radiation (0.1 and 3 Gy) increased OS and vascular endothelial growth factor (VEGF) expression, with an additive effect when the highest dose was combined with ID. However, in MCF7 cells no additive effect was observed. VEGF mRNA up-regulation was reactive oxygen species (ROS)-dependent, involving radiation-induced mitochondrial ROS. Results on total VEGF mRNA hold true for the pro-angiogenic isoform VEGF165 mRNA, but the treatments did not modulate the anti-angiogenic isoform VEGF165b. Radiation-induced antioxidant response was differentially regulated upon ID in both cell lines. Thus, radiation response is modulated according to iodine status and cell type and can lead to additive effects on ROS and VEGF. As these are often involved in cancer initiation and progression, we believe that iodine status should be taken into account in radiation prevention policies.

## 1. Introduction

The human population is constantly exposed to diverse types of radiation through natural sources such as UV, radioactive materials from earth or cosmic rays; or through man-made sources mainly from medical exposure [1,2]. Breast is a radiation-sensitive organ, as demonstrated by several epidemiological studies showing a link between radiation exposure and breast cancer risk [3,4,5]. For a better understanding of breast cancer risks associated with radiation exposure, studies on the molecular targets of radiations are essential. Some genes are often up regulated by radiation exposure, such as the vascular endothelial growth factor (VEGF) gene, involved in vascularization and angiogenesis. Indeed, several studies demonstrated radiation-induced increases in VEGF expression in vivo and in vitro in diverse organs and cell lines, among which were breast cell lines [6,7,8]. Due to the important role of angiogenesis in tumor progression, but also in vasculature protection after radiation exposure, several authors have hypothesized a role for VEGF in radiation resistance [8,9,10,11]. A likely explanation for this VEGF up-regulation involves radiation-induced reactive oxygen species (ROS). Indeed, ionizing radiation is known to increase ROS production through different mechanisms, such as radiolysis of water, mitochondrial disruption or the activation of ROS-producing enzymes [2,12,13,14], and VEGF expression is known to be regulated by ROS content and oxidative stress (OS) via, for instance, ROS- and oxidative stress (OS)-sensitive transcription factors [15,16]. However, radiation exposure is also often followed by an enhancement of the antioxidant defenses. By increasing either the activity or the expression of antioxidant enzymes, cells try to regulate the sudden burst in cellular ROS and their consequences, thereby potentially leading to radiation resistance [17,18,19]. However, it is important to note that the effects of ionizing radiation on human health and cancer risk vary according to the dose and type of radiation used [20,21]. Currently, the biological effects of low doses of radiation are mostly extrapolated from the effects of the high doses, based on the linear non-threshold model. But increasing results tend to disagree with this model, and fundamental research is thus needed to assess low dose radiation effects [20,22,23,24].

Another factor that can influence breast pathologies is iodine deficiency (ID). ID was shown to induce breast atypia [25,26,27], and a lower breast cancer rate was observed in various populations known for their high iodine intake [28,29,30]. Moreover, acute ID was shown to induce a transient ROS-dependent VEGF overexpression in vivo and in vitro in different organs capable of iodine uptake such as thyroid, salivary glands, stomach, and mammary glands [31,32,33]. Despite ongoing efforts to improve iodide supply, ID prevalence is still a huge concern, with 19 countries considered iodide-deficient Moreover, various sub-populations in iodide-sufficient countries have insufficient iodide intake due to their diet choice or to limited access to sea products [34]. Therefore, ID’s potential interference with the radiation response is an interesting question. In this regard, iodine intake was previously found to modulate radiation effects in thyroid. It was indeed observed that a long-term iodine deficient diet increases X-radiation induced carcinogenesis in rats [35]. Since several studies have linked iodine to breast health and because breast is known as radiation-sensitive, a potential additive effect of these two factors in this organ is also conceivable. 

Hence, as radiation and ID are able to increase ROS content and up-regulate VEGF expression, we hypothesized that the combination of both factors could lead to an additive effect regarding ROS production and VEGF regulation in breast cells, and that ID could influence breast response to radiation exposure. In this study, the molecular mechanisms behind the oxidative stress induced by radiation alone or in combination with ID were studied in two Na+/I- symporter (NIS)-expressing breast cell lines already recognized as ID-sensitive: a cancerous (MCF7) and a non-cancerous (MCF12A) one. 

## 2. Results

### 2.1. Additive Effects of Radiation and ID on Total VEGF and VEGF165 mRNA Up-Regulation in MCF12A But Not in MCF7 Cells

Two cell lines known to express the sodium/iodide symporter NIS and to be influenced by acute ID [33] were used. MCF12A and MCF7 cells were exposed to X-ray doses of 0.05, 0.1, and 3 Gy 30 min after medium change. Cells were harvested 3, 4, 6, or 24 h after medium change and thus 2.5 h, 3.5 h, 5.5 h, or 23.5 h after irradiation. Short experimental durations were used because ID was previously shown to increase VEGF from 3 to 6 h after medium change in these cell lines [33]. In iodine-sufficient conditions, the two highest radiation doses (0.1 and 3 Gy) induced a similar increase in VEGF mRNA expression in MCF12A cells at 4 h (Figure 1), but not in MCF7 cells, while the 0.05 Gy dose had no effect in both cell lines. Cells were then exposed to acute ID alone, radiation alone or acute ID and radiation (Figure 2). Acute ID induced VEGF mRNA up-regulation after 4 h in MCF12A (Figure 2A–C) and after 6 h in MCF7 cells (Figure 2D–F), which is in agreement with our previous results [33]. Interestingly, the combination of ID with 3 Gy X-rays had an additive effect on VEGF mRNA in MCF12A cells (Figure 2C), but showed a similar VEGF mRNA induction to that induced by ID alone in MCF7 cells at all the time points studied (Figure 2F). No additive effect on VEGF expression was observed while using lower doses in both cell lines (Figure 2A,B,D,E), even though exposure to 0.1 Gy of X-rays in both iodine-sufficient and iodine-deficient conditions led to a VEGF increase. VEGF up-regulation was transient as none of the above-mentioned effect was observed after 24 h.

The expression of VEGF protein was then visualized by Western blot in cells exposed to ID and/or radiation doses of either 0.1 or 3 Gy, as these were the doses which increased VEGF mRNA expression in at least one of the cell lines. For protein content analysis, cells were harvested one hour later compared to mRNA analysis. The observation made on VEGF mRNA expression held true for the VEGF protein expression in both cell lines (Figure 3). 

VEGF pre-RNA is known to be alternatively spliced and splice variants have different angiogenic capacities [36]. One of the principal splice variants involved in pro-angiogenic response is VEGF165 [37]. We have therefore looked at its mRNA expression to verify whether an additive effect was also observed on this isoform when cells were simultaneously exposed to ID and 3 Gy X-rays in MCF12A cells. Although no additive effect was observed in MCF7 cells, the VEGF165 isoform was also looked at in this cell line to verify whether the treatment had the same effects. A similar response was observed on VEGF165 mRNA and on global VEGF mRNA regulation in both cell lines (Figure 4A,B). The alternate splicing of VEGF exon 8 leads to splice variants that do not have pro-angiogenic activity and that are currently considered as competitive inhibitors of the pro-angiogenic isoforms. They are termed as VEGFxxxb [38]. We have therefore looked at the negative splice variant VEGF165b to see whether it is also up-regulated. ID and 3 Gy of X-rays separately or combined did not significantly modulate VEGF165b mRNA in both cell lines (Figure 4C,D). 

### 2.2. Radiation- and ID-Induced ROS Production and Oxidative Stress in MCF12A and MCF7 Cells

Because VEGF is known to be regulated by ROS and because ROS were previously shown to be involved in ID-induced VEGF up-regulation in breast cells and other organs [33,39], ROS production was indirectly assessed through 4-HNE staining, a product of lipid peroxidation and a marker of oxidative stress, and directly via the use of DCF-DA dye. 4-HNE staining revealed an increase in oxidative stress 2 and 4 h after ID induction or irradiation (0.1 and 3 Gy) in MCF12A (Figure 5A–F) and MCF7 cells (Figure 5G–L) respectively. However, the increase in HNE staining was visually stronger in MCF12A cells exposed to ID and 3 Gy of X-rays simultaneously (Figure 5F), but this additive effect was not observed in MCF7 cells (Figure 5L). 

Since the additive effect of ID and radiation on VEGF mRNA and HNE staining was only observed in MCF12A at a dose of 3 Gy, ROS induction was further assessed in those cells. ROS content was detected with DCF-DA dye and quantified by flow cytometry. ID was induced by medium change and lasted 1 or 2 h. ID alone induced a moderate increase in ROS at both time points (Figure 6A) which is in agreement with previous observations [33]. Control and ID cells were irradiated after medium change and cells were exposed to the dye 10 min before irradiation (Figure 6B), or 25 min after the end of irradiation (Figure 6C). When the dye was added before the irradiation, a strong increase in ROS content was observed, probably due to water radiolysis, (Figure 6B). When the dye was added 25 min after irradiation, no ROS increase was observed in cells exposed to X-rays alone (Figure 6C) which is consistent with observations in the literature describing a short burst of ROS following irradiation which disappears within seconds [2,12]. However, no significant difference in ROS content was observed between irradiated iodine-sufficient and irradiated iodine-deprived cells when the dye was added before the irradiation (Figure 6B). Moreover, the ROS production in ID cells and in ID irradiated cells was increased to a similar extent when the dye was added 25 min after irradiation (Figure 6C). Thus, our data suggest that ID induces a modest increase in ROS that lasts until two hours of treatment which is accompanied by a strong increase in ROS induced by 3 Gy X-rays exposure, but which lasted for a short time.

### 2.3. ID and Radiation-Induced Increase in VEGF mRNA Depends on Cellular ROS

We previously demonstrated with the use of the antioxidant *N*-acetylcysteine (NAC) that ID-induced VEGF up-regulation was dependent on ROS in both cell lines [33]. To verify ROS involvement in radiation-induced VEGF up-regulation, MCF7 and MCF12A cells exposed to 3 Gy X-rays with or without ID were treated with NAC (Appendix A). VEGF up-regulation was no longer observed after the treatment in presence of NAC, in both cell lines. 

Two distinct inhibitors of cellular ROS were then used to shed some light on the origin of ROS in our model. Firstly, to know whether ROS from cellular production are involved in VEGF mRNA regulation, its expression was assessed in presence of DPI, an inhibitor that targets flavoenzymes, a major cellular source of ROS, and that partially targets mitochondrial ROS [40,41]. VEGF mRNA increase induced by ID and radiation alone or in combination in MCF12A cells was totally inhibited by DPI (Figure 7A). Likewise, VEGF mRNA increase induced by ID and by ID combined with radiations was not observed anymore in MCF7 cells treated with DPI (Figure 7B). Flavoenzymes are thus involved in the ROS-induced VEGF up-regulation that is observed with the different treatments in both cell lines.

Another important source of ROS is the mitochondria, particularly through mitochondrial electron transport superoxide leakage [41]. Thus, in a second time, VEGF mRNA was studied in presence of mitoTEMPO, a mitochondrial superoxide and alkyl radical scavenger. In MCF12A cells, mitoTEMPO did not prevent the ID-induced increase in VEGF mRNA. However, it prevented the increase of VEGF mRNA induced by radiation regardless of the Iodide status (Figure 7C). Likewise, the presence of mitoTEMPO did not prevent ID-induced VEGF mRNA increase in MCF7. MitoTEMPO did not block the increase induced by the co-treatment either (Figure 7D). Mitochondrial ROS are thus involved in radiation-induced VEGF up-regulation observed in MCF12A cells. 

### 2.4. Additive Up-Regulation of Antioxidant Enzymes mRNA by ID and Radiation 

Because ID and both the 0.1 and 3 Gy doses of X-rays could induce oxidative stress, the mRNA of three important antioxidant enzymes, superoxide dismutase (SOD) 1 and 2 and catalase, was assessed in both cell lines exposed to ID and/or 0.1 or 3 Gy X-rays (Figure 8).

In MCF12A cells, SOD2 mRNA was only upregulated when cells were exposed to both ID and 0.1 Gy (Figure 8A,B) of X-rays while SOD1 mRNA was upregulated in cells exposed to both ID and 0.1 or 3 Gy of X-rays (Figure 8C,D). None of the treatments influenced catalase mRNA expression at the time points studied (Figure 8E,F). In MCF7 cells, similar observations were made regarding SOD2 mRNA expression as an additive effect was detected in iodine-deficient cells irradiated only at 0.1 Gy (Figure 8G,H). None of the treatments significantly affected SOD1 and catalase mRNA expression at the time points studied (Figure 8I–L).

## 3. Discussion

Ionizing radiation and ID are both known to influence breast pathologies and to elicit changes in molecular pathways, such as the activation of pathways regulating VEGF expression [5,6,8,30,33]. Here we report that, in some conditions, the combination of these two factors may induce greater VEGF up-regulation than when applied separately. In this research, two breast cell lines known for their sensitivity to ID were used. However, both cell lines reacted differently to radiation with or without ID. In MCF12A cells, a single X-ray dose of 3 Gy increased VEGF mRNA expression similarly to acute ID, and their effects on VEGF expression were additive. This VEGF up-regulation was accompanied by an increased oxidative stress that followed the same pattern. However, in MCF7 cells irradiation did not affect VEGF expression. The combination of X-rays with ID up-regulated VEGF mRNA expression to an extent similar to that of acute ID alone. Nevertheless, OS was slightly increased by the different treatments, i.e., ID, 3 or 0.1 Gy X-Rays, and the combination of ID and irradiation. Thus, radiation-induced OS under iodide-sufficient conditions did not affect VEGF mRNA expression at the studied time points and it is likely that the effects observed on VEGF expression in irradiated ID cells were only due to acute ID. Though both ID and radiation induced an increase in OS in MCF7 cells, the origin and type of ROS might be different in both conditions, affecting thus different molecular pathways in these cells, thereby leading to increased VEGF expression in one case while showing no effect on VEGF in the other case. ROS exact chemical nature and origin in ID and in radiation-exposed cells could be interesting to characterize as it could lead to a better understanding of VEGF regulation by ROS. Of note, another group observed that electromagnetic radiation (γ rays, 2 Gy) induced VEGF up-regulation in MCF7 cells after 16 h [7]. Though the radiation type and dose were different, it is thus possible that radiation could induce VEGF mRNA up-regulation after longer experimental durations. However, we did not observe any changes in VEGF mRNA expression 24 h after irradiation either. Several studies have shown that doses over 1 Gy induce VEGF up-regulation in various cell types or tissues [6,7,11,42]. Here we showed an increase of VEGF mRNA expression starting from 0.1 Gy under both iodide-sufficient and -deficient conditions in MCF12A cells. However, in our model, the VEGF overexpression induced by both doses of radiation does not seem to be dose-dependent but rather seems to be induced by a threshold dose reached after exposure to a 0.1 Gy dose of X-rays.

Because MCF7 cells are cancerous and MCF12A are not, an interesting question to be raised is whether the differential VEGF regulation reflects the physiological state of those cells. Indeed, iodide transporters (NIS and pendrin) regulation was described to vary in normal, cancerous or lactating glands and to be hormonally dependent [43,44,45,46]. It is thus likely that radiation response modulation by iodide is related to the gland pathophysiology. Moreover, a higher OS level is often described in breast cancer cell lines, including MCF7 cells [47]. Cancerous cell lines could be more tolerant to certain ROS types increase as the ROS overload is less than in a non-cancerous cell line. In this regard, though the trends were similar between both cell lines, the antioxidant enzyme activation in MCF12A cells was more significant than in MCF7, potentially reflecting a greater stress due to the ROS increase in these cells. Mitochondria morphology was also described to be affected in breast cancer cell lines compared to non-tumorigenic cell lines, including in MCF7 cells [48]. As ROS induced by a high dose of radiation were shown here to originate at least partially mitochondrial ROS, it could be hypothesized that this difference in mitochondrial morphology is related to the difference in the results observed between the two cell lines. These questions require further investigation. Moreover, since such a different response is observed in both cell lines, it could be interesting to extend this study to other cell lines. In this regard, different breast cancer cell lines were recently described to express NIS, such as SKBR3, T47D, MDA-MB-231, and MDA-MB-468 [49,50]. Moreover, different subtypes of breast cancer have been described and the reaction to current treatment may vary according to the subtype [51]. Extending our research to the above-mentioned cell lines would cover a larger amount of breast cancer subtypes and lead to a better understanding of the influence of iodine status on radiation sensitivity. Regarding non-cancerous cell lines, very few data are available on their NIS expression. However, if necessary, NIS expression can be stimulated by the combination of estrogen, prolactin, and oxytocin, since these hormones have been shown to induce NIS expression in vivo, or by using tRA or EGF combined with ATP, according to their receptor status, as described in different breast cell lines [43,49,52]. 

Ionizing radiation, among which X-rays, were shown to affect alternative splicing in various cell and tissue types including breast [53,54]. In our study, radiations were found to differentially influence the expression of VEGF alternative transcripts under iodide-deficient and -sufficient conditions. Overall, the expression of VEGF165 mRNA followed the same trend as the total VEGF mRNA, while that of VEGF165b mRNA was not affected by the treatments. Because VEGFxxxb splice variants are hypothesized to competitively inhibit the pro-angiogenic effects of VEGFxxx splice variants and because VEGF165 is one of the principal pro-angiogenic variants of VEGF [36,55], this result suggests that the up-regulation of VEGF mRNA is likely to induce a microvascular or angiogenic response. Further investigation is thus necessary to confirm this hypothesis. Interestingly, acute ID also induced VEGF165 mRNA up-regulation while it had no effect on that of VEGF165b. To the best of our knowledge, this is the first study reporting that iodide status can influence alternative splicing and an ID-induced upregulation of pro-angiogenic isoforms is in agreement with previous results showing a VEGF-dependent increased blood perfusion induced by ID in mice [33]. 

Because a difference in OS induced by the exposure to 3 Gy of X-rays was visually observed in iodide-deficient and iodide-sufficient MCF12A cells and because ROS and OS play an important role in VEGF induction, ROS production was quantified by flow cytometry. Acute ID alone induced a modest increase in ROS at one and two hours after exposure. This modest increase is sufficient to induce VEGF as NAC treatment abrogates ID-induced VEGF [33]. As to radiation exposure, it induces a rapid strong ROS increase that could not be observed anymore 25 min after the end of the irradiation. This observation is in accordance with the literature where a sharp short-lasting rise, probably due to water radiolysis, is usually described before a second rise several hours later according to the cell type [1,12]. This radiation-induced strong rise in ROS content was not significantly influenced by acute ID. However, irradiated iodide-deficient cells are exposed to two subsequent rises in ROS, which may account for the additive effects of the treatments on oxidative stress. These two additional sources of ROS are probably also responsible for the observed higher VEGF mRNA up-regulation since ROS and OS are known to be involved in VEGF regulation and ROS quenching with the broad-scaled ROS inhibitor NAC prevented VEGF mRNA up-regulation. It is possible that ROS induced by ID and/or radiation act either through the activation of the same pathway, or via different mechanisms whose effects could be additional. 

We then investigated the source of ROS involved in VEGF up-regulation in both cell lines. To verify whether other ROS than those resulting from water radiolysis were involved in VEGF regulation, DPI was used to inhibit flavoenzymes, a major ROS cellular source. Flavoenzymes are mostly represented by NADPH oxidases but also include enzymes such as xanthine oxidase and NADPH cytochrome P450 oxidoreductase. DPI also partially inhibits mitochondrial ROS, another important ROS source [56]. Then, mitochondrial superoxide, which mainly comes from mitochondrial electron chain transport leaks but could also partly originate from enzymes such as NOX4 which was suggested to be expressed in breast mitochondria [41,57], and mitochondrial alkyl radicals were inhibited by mitoTEMPO. In MCF12A cells, acute ID induced VEGF up-regulation was found to be dependent on ROS from cellular activity, but not on mitochondrial superoxide and alkyl radicals. VEGF overexpression induced by 3 Gy of X-rays in iodide-deficient and iodide-sufficient conditions however was decreased by both DPI and mitoTEMPO, suggesting the involvement of ROS from mitochondrial origin. However, those results do not exclude a likely participation of ROS from radiolysis of water, and maybe from other cellular sources. Nevertheless, it is interesting to note that VEGF mRNA up-regulation induced by radiation is not only dependent on ROS induced by water radiolysis, but also depends, at least partially, on intrinsic cellular ROS production. However, our data suggest that ROS induced by ID and radiation have different cellular origin, and the addition of these different sources of ROS, as visualized by flow cytometry, might account for the additive effects observed on VEGF expression. 

Though the induction of VEGF mRNA was not higher in cells exposed to X-rays combined to acute ID as compared to ID cells alone in MCF7 cell line, ROS were inhibited with the same inhibitor to see whether the source of ROS was the same in both treatments. VEGF up-regulation was found to be inhibited by DPI but not by mitoTEMPO in both treatments, suggesting that it depends on ROS resulting from cellular activity but not on mitochondrial superoxide and alkyl radicals. 

Besides, the combination of ID and radiation exposure led to the up-regulation of antioxidant enzymes mRNA in both cell lines, though, despite trends towards increases, the exposure to only one of the factors did not significantly up-regulate them at the time-points studied. Our data suggests that the general response on antioxidant enzymes is stronger in iodide-deficient cells exposed to 0.1 Gy than those exposed to 3 Gy at the experimental durations studied, as more enzymes are up-regulated at 0.1 than at 3 Gy. As observed in other biological models, activation of cellular defense mechanisms might contribute to survival and radiation resistance [17,18,19]. Though the involvement of the antioxidant system was not assessed in our model, it is possible that a stronger antioxidant defense activation after exposure to 0.1 Gy compared to the one induced after exposure to 3 Gy in ID MCF12A cells is responsible for the lack of additive effects between 0.1 Gy of X-rays and ID in these cells. Yet, it should be kept in mind that a regulation of these enzymes at later timing or at the protein level is also conceivable in the conditions where no mRNA up-regulation was observed. It is also interesting to note that SOD2 mRNA expression is up regulated in ID MCF12A cells exposed to 0.1 Gy but not to 3 Gy at the time point studied. As this isoform is expressed in the mitochondria and VEGF overexpression was inhibited by mitoTEMPO in ID MCF12A cells exposed to 3 Gy but not in ID cells, a possible involvement of this enzyme in radiation-induced ROS regulation could be imagined. However, mitochondrial ROS involvement in cells exposed to 0.1 Gy of X-rays should first be verified, as was likely the case in cells exposed to 3 Gy. Moreover, VEGF-induced up-regulation in both cell lines is transient, as it was not observed anymore after 24 h. Antioxidant enzymes’ involvement is possible in this regulation in cells exposed to ID and radiation and further investigation is needed to verify this hypothesis.

Overall, radiation doses of 0.1 and 3 Gy increase OS in both the cancerous and the non-cancerous cell line and up-regulate VEGF expression in the non-cancerous cell line, MCF12A, leading to ROS-dependent additive effects on VEGF expression when the high dose is applied in combination with ID in the same cell line. In addition, antioxidant enzymes expression is also modulated following radiation and iodine deficiency with a general tendency towards a stronger up-regulation with the co-treatment, particularly in the case of the 0.1 Gy dose exposure. This up-regulation of antioxidant enzymes might have an effect on VEGF overexpression regulation and needs further investigation. 

During radiotherapy, healthy cells in the direct vicinity of cancer cells can be exposed to doses such as 0.1 Gy and higher and could thus secrete VEGF, thereby protecting the tumor vasculature. they might escape radiation-induced apoptosis through the up-regulation of antioxidant enzymes, which are two mechanisms often described in cancer radioresistance [8,9,10,11,18,19]. Moreover, 0.1 Gy can also be reached via repeated medical exposures to ionizing radiation such as CT-scans [58]. Therefore, the effect of repeated exposure on VEGF/antioxidant enzymes expression needs further research since their up-regulation could have deleterious consequences in cancer or pre-cancer tissues [59,60,61]. Here we also showed that a high dose of radiation that is relevant to radiotherapy [62,63] induces a higher up-regulation of VEGF expression and/or some antioxidant mechanisms under iodide-deficient conditions. Our data suggest that iodide status of patients undergoing radiotherapy should be taken into account before exposure to ionizing radiation as it could potentially modulate the radiotherapy outcome through its influence on the tumoral micro-environment.

## 4. Materials and Methods 

### 4.1. Cell Models

MCF7 human mammary gland adenocarcinoma cell line (Sigma, St.-Louis, MO, USA) and MCF12A human normal mammary gland cell line (ATCC, Manassas, VA, USA) were routinely cultured at 5% CO_2_, 37 °C in a humidified atmosphere, as previously described [33]. Seven days before the experiments, 10^–8^ M NaI was added to the medium. On the day of the experiment, 0.75 mM *N*-acetylcysteine (NAC, Sigma) was added to the medium 1 h before replacing the medium where indicated. After washing with PBS, culture medium was replaced by fresh medium containing NaI (controls) or not (ID) with or without 10 µM diphenyleneiodonium (DPI) (Sigma), 0.75 mM NAC, or 10 µM (MCF7 cells) or 2 µM (MCF12A cells) mitoTEMPO (Sigma) where indicated. Cells were harvested 1 to 6 h after medium renewal, since ROS and VEGF mRNA are known to increase at these time points during ID in these cell lines [33], and after 24 h.

### 4.2. Irradiation

30 min after medium change, cells were X-irradiated once with a dose of 0.05, 0.1, or 3 Gy, using an Xstrahl RX generator (operating at 250 kV and12 mA, 3.8 mm Al-, and 1.4 mm Cu-filtered X-rays). 

### 4.3. Quantitative Polymerase Chain Reaction

Cell were harvested and homogenized in 1 mL TriPure isolation reagent (Roche, Mannheim, Germany), and total RNA was extracted according to the manufacturer’s protocol and resuspended in RNase-free water. Reverse transcription was performed as previously described [31]. cDNAs (0.04 µg of cDNA template in 2 µL) were mixed with 10 µM of selected primer pair (Table 1) and Perfecta SYBR green reaction mix (VWR, Radnor, PA, USA) in a final volume of 15 µL. Reactions were performed with an IQ5 iCycler (Bio-Rad, Herts, UK) as previously described [64]. Annealing temperatures are given in Table 1. Amplification levels were normalized to those of β-actin. All samples were measured in duplicate.

### 4.4. Western Blot

Cells were suspended into Laemmli buffer (10 mL Glycerol, 2 g SDS, 0.756 g Tris-hydroxymethylaminomethan, 0.19 g EDTA (5 mM) in a 100 mL distilled water solution) containing protease inhibitors (protease inhibitor cocktail tablets, Roche, Mannheim, Germany) and sonicated for 15 s. The protein concentration was measured with a BCA Protein assay kit (Pierce, Rockford, IL, USA) according to manufacturer’s protocol. Western blotting was performed as previously described [65]. VEGF (1 µg/mL; ab46154, Abcam, Cambridge, UK) or GAPDH (1/5000; 2118S, Cell signaling, Danvers, MA, USA) primary antibody was incubated overnight at 4 °C and the secondary antibody (1:5000; #31460, Thermo Fisher Scientific, Waltham, MA, USA) was incubated at room temperature for one hour. Signal was detected using the ECL Western blotting detection reagent (VWR). Protein expression was quantified by densitometry using NIH Scion Image Analysis Software (National Institutes of Health, Bethesda, Rockville Pike, MA, USA). VEGF expression was normalized to GAPDH expression. 

### 4.5. Immunofluorescence

Immunofluorescence was performed as previously described [33]. Plates were incubated at 4 °C overnight with a primary antibody (4-hydroxynonenal (4-HNE), 393207-100, VWR) at a concentration of 1/1000. The primary antibody was omitted in negative controls. Signals were revealed after 1 h incubation at room temperature in the dark with a species-specific secondary antibody conjugated to fluorophores (Alexa Fluor 488-conjugated goat anti-rabbit, A11034, 1/300, Invitrogen, Ghent, Belgium). Nuclei were stained for 5 min with 4′-6′diamidino-2-phenylindole (DAPI, 1/10,000) and slides were mounted in Fluorescent Mounting Medium (Dako Cytomation, Heverlee, Belgium). Images were captured on an AxioCam MRc5 fluorescence microscope using the Axio Vision 4.8 software (Zeiss, Zaventem, Belgium).

### 4.6. ROS Measurement Using Flow Cytometry

On the day of the experiment, medium change was executed with phenol red free medium. ROS production was measured with a fluorescent dye Carboxy-DCFDA (5-(and-6)-Carboxy-2’,7’-Dichlorofluorescein Diacetate) (DCF-DA, Invitrogen). Cells were washed with PBS and incubated at 37 °C, 5% CO_2_ during 45 min with the DCF-DA dye at a final concentration of 10 µM in HEPES/HBSS with Ca^2+^/Mg^2+^ (Thermo Scientific, Ghent, Belgium). Cells were washed and trypsinized. Harvested cells were centrifuged at 200× *g* for 5 min and supernatant was discarded. Pellets were washed and resuspended in HBSS without Ca^2+^/Mg^2+^. Cells were filtered to avoid aggregates and propidium iodide was added at a final concentration of 1 µg/mL to eliminate dead cells from the measurements. Positive ROS control was obtained by adding a ROS inducer, tert-butyl hydroxide, at 25 or 50 µM in untreated cells. DCFDA and propidium iodide fluorescence were measured with a BD Accuri C6 flow cytometer (BD biosciences, New Jersey, NJ, USA) at 525 and 620 nm respectively. Maximum flow speed was 300 events/s and a total of 10 000 events per sample were analyzed.

To verify that radiation does not affect the dye when it was added before radiation exposure, we exposed the de-esterified DFC-DA dye in a PBS solution without cells to 3 Gy of X-rays in presence and absence of NAC. An increase in fluorescence was induced by the exposure of the de-esterified dye to 3 Gy of X-rays, but this increase was not observed in presence of NAC, suggesting that it was only due to ROS production following water radiolysis and not the sensitivity of the dye towards radiation exposure. Similar results were previously obtained by Rappole et al. (2012) [66].

### 4.7. Statistics

All data are expressed as means ± SEM. Experiments were independently conducted 3 to 8 times with 3 to 5 technical replicates for each condition. One-way ANOVA with Tukey, Dunnett or Newman-Keuls post-hoc test were used where appropriate (GraphPad Instat, San Diego, CA, USA). Statistical details are given in the legend of each figure. *p* < 0.05 was considered to be statistically significant.

## Figures and Tables

**Figure 1 ijms-21-03963-f001:**
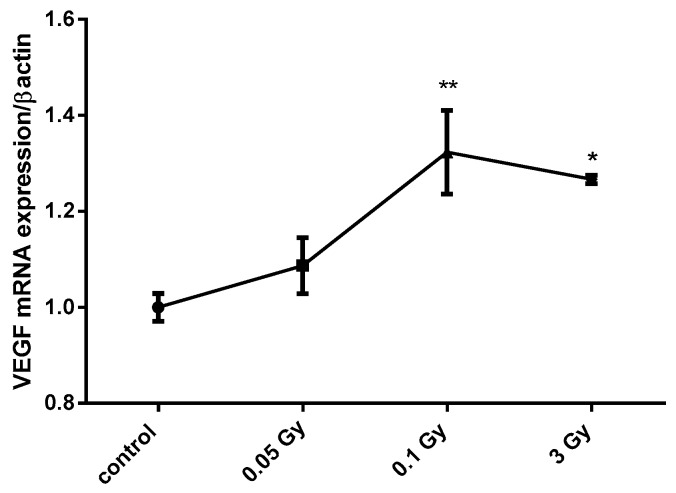
X-ray doses from 0.1 to 3 Gy increase vascular endothelial growth factor (VEGF) mRNA expression in MCF12A breast cells. MCF12A cells were cultured in iodine-containing medium for 7 days. Thereafter, the medium was replaced by fresh iodine-containing medium and cells were then irradiated with a dose of 0.05, 0.1, or 3 Gy. Cells were harvested 4 h after medium change. VEGF mRNA expression was determined using RT-qPCR. Data are expressed as means ± SEM. *p*-values < 0.05 were considered as statistically significant. **p* < 0.05, ***p* < 0.01. Statistical test: one-way ANOVA with Dunnett post-hoc test. *N* = 3, one representative experiment.

**Figure 2 ijms-21-03963-f002:**
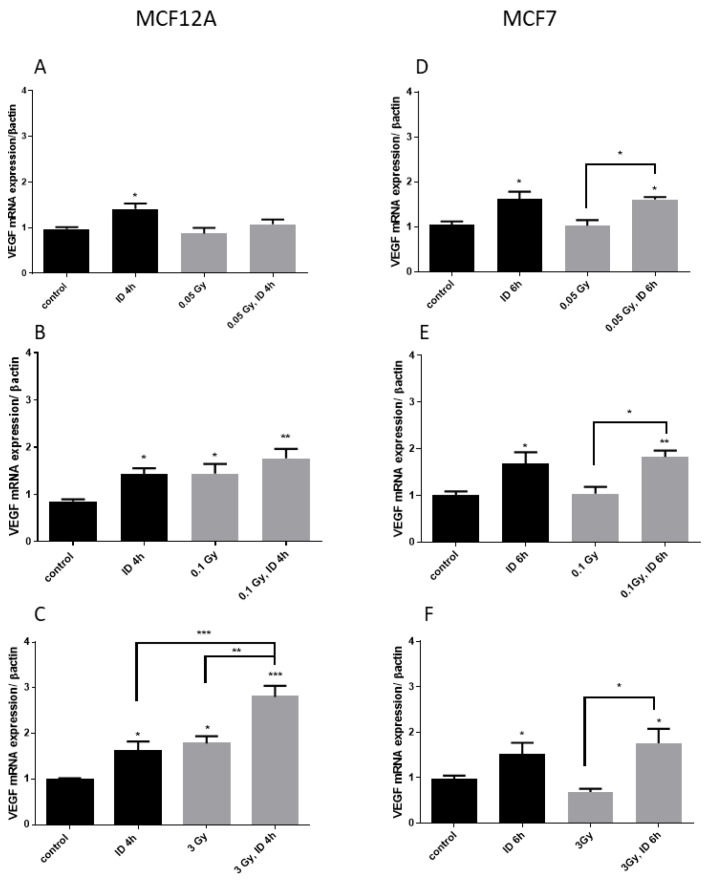
Iodine deficiency (ID) and 3 Gy X-rays additively up-regulate VEGF mRNA in MCF12A but not in MCF7 cells. Cells were cultured in iodine-containing medium for 7 days. Thereafter, the medium was replaced by iodine-containing (control) or iodine deficient (ID) medium and cells were then irradiated with a dose of 0.05 (**A**,**D**), 0.1 (**B**,**E**), or 3 Gy (**C**,**F**). Cells were harvested 4 or 6 h after medium change. VEGF mRNA expression in MCF12A (A–C) and MCF7 cells (D–F) was determined using RT-qPCR. Data are expressed as means ± SEM. *p*-values < 0.05 were considered as statistically significant. **p* < 0.05, ***p* < 0.01, ****p* < 0.001. Statistical test: one-way ANOVA with Tukey post-hoc test. A, B, F: *N* = 5; C: *N* = 7; D: *N* = 3; E: *N* = 4.

**Figure 3 ijms-21-03963-f003:**
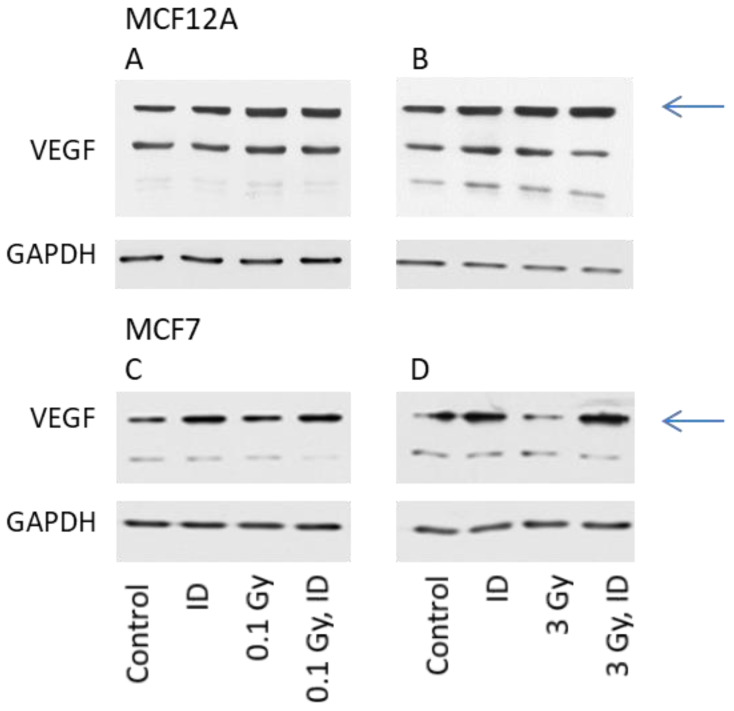
ID and radiation differentially up-regulate VEGF protein in MCF12A and MCF7 breast cells. MCF12A (**A**,**B**) and MCF7 (**C**,**D**) cells were cultured in iodine-containing medium for 7 days. Thereafter, the medium was replaced by iodine-containing (control) or iodine deficient (ID) medium and cells were irradiated with a dose of 0.1 (A, C) or 3 Gy (B, D). Cells were harvested 5 (A, B) or 7 (C, D) hours after medium change. VEGF protein expression was visualized by western blot. *N* = 3.

**Figure 4 ijms-21-03963-f004:**
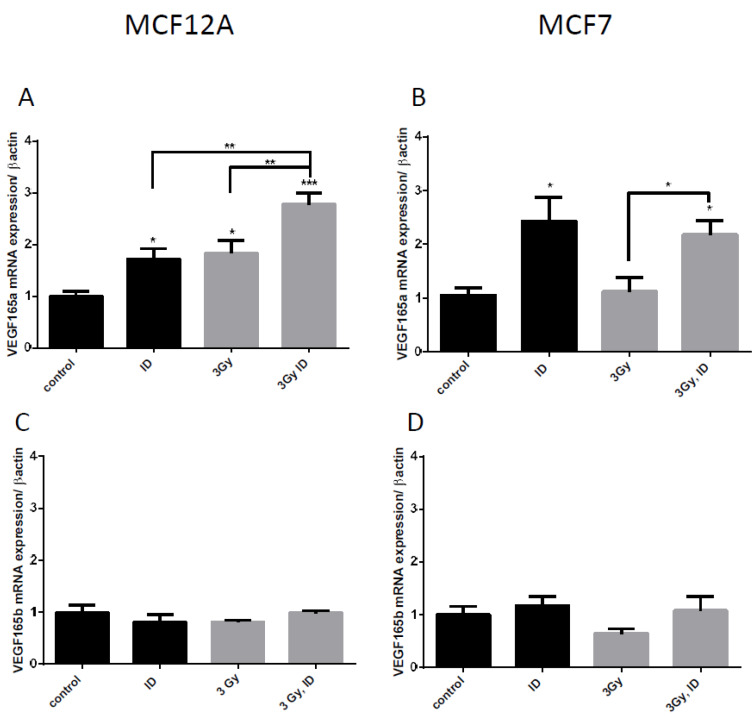
ID and irradiation modifies VEGF alternative splicing in breast cells. Cells were cultured in iodine-containing medium for 7 days. Thereafter, the medium was replaced by iodine-containing (control) or iodine deficient (ID) medium and cells were then irradiated with a dose of 0.1 or 3 Gy. Cells were harvested 4 (**A**,**C**) or 6 h (**B**,**D**) after medium change. VEGF 165 (A, B) and VEGF165b (C, D) mRNA expression in MCF12A (A, C) and MCF7 cells (B, D) was determined using RT-qPCR. Data are expressed as means ± SEM. *p*-values < 0.05 were considered as statistically significant. **p* < 0.05, ***p* < 0.01, ****p* < 0.001. Statistical test: one-way ANOVA with Tukey post-hoc test. A: *N* = 5; B: *N* = 4; C–D: *N* = 3.

**Figure 5 ijms-21-03963-f005:**
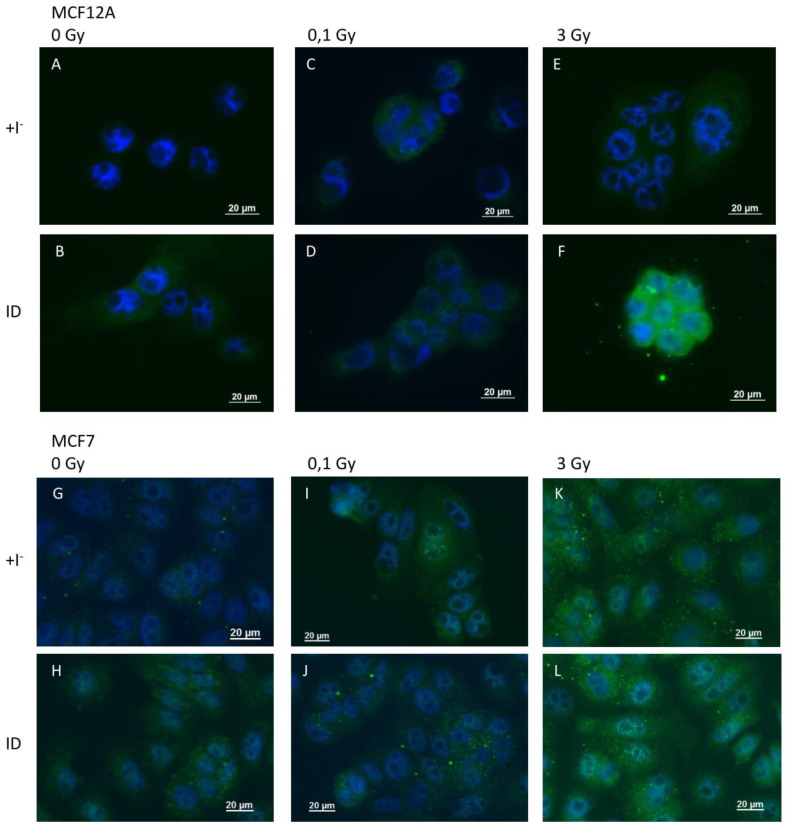
ID and X-irradiation increase oxidative stress (OS) in breast cells, with an additive effect at 3 Gy in MCF12A cells. Cells were cultured in iodine-containing medium for 7 days. Thereafter, the medium was replaced by iodine-containing (control) or iodine deficient (ID) medium and cells were then irradiated with a dose of 0.1 or 3 Gy. 4-HNE adducts were detected by immunofluorescence after 2 and 4 h in MCF12A (**A**–**F**) and MCF7 (**G**–**L**) cells respectively. Control (A, G), ID (B, H), 0.1 Gy (C, I), ID + 0.1 gy (D, J), 3 Gy X-rays (E, K) and ID + 3 Gy X-rays (F, L). Representative pictures are shown. Scale bars = 20 µm. *N* = 3.

**Figure 6 ijms-21-03963-f006:**
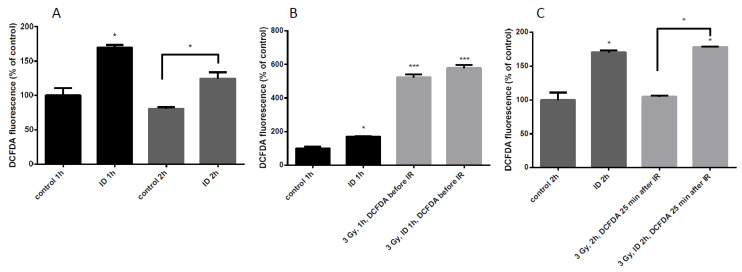
ID and radiation differentially increase ROS cellular content in MCF12A cells. Cells were cultured in iodine-containing medium for 7 days. Thereafter, the medium was replaced by iodine-containing (control) or iodine deficient (ID) medium and cells were irradiated with a dose of 3 Gy. Cells were harvested 1 (**A** (two first columns), **B**) or 2 h (A (two last columns), and **C**) after medium change. Cellular reactive oxygen species (ROS) content was visualized with DCF-DA and measured by flow cytometry. In irradiated cells, DCF-DA was added 10 min before exposure to radiation (B) or 25 min after the irradiation (C). DCF-DA was added at corresponding timing in non-irradiated cells. Data are expressed as means ± SEM. *p*-values < 0.05 were considered as statistically significant. **p* < 0.05, ****p* < 0.001. Statistical test: one-way ANOVA with Tukey post-hoc test. *N* = 3, one representative experiment.

**Figure 7 ijms-21-03963-f007:**
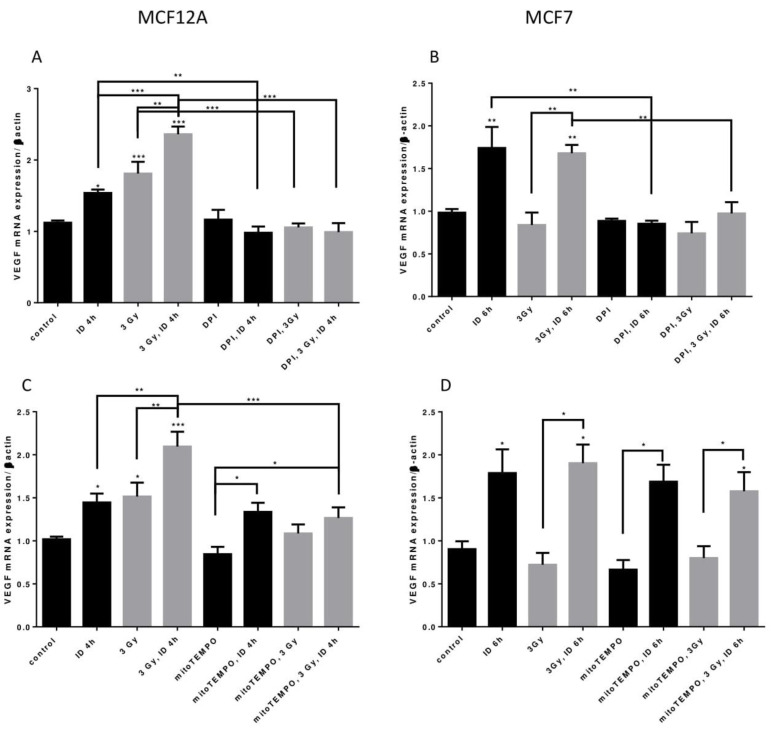
ID and radiations-induced VEGF mRNA up-regulation depends on cellular ROS production from different origins. Cells were cultured in iodine-containing medium for 7 days. Medium was replaced by iodine-containing (control) or iodine deficient (ID) medium and cells were then irradiated with a dose of 3 Gy. Part of the cells was treated with DPI (**A**,**B**) or mitoTEMPO (**C**,**D**). Cells were harvested 4 or 6 h after medium change. VEGF mRNA expression in MCF12A (A, C) and MCF7 cells (B, D) was determined using RT-qPCR. Data are expressed as means ± SEM. *p*-value < 0.05 were considered as statistically significant. **p* < 0.05, ***p* < 0.01, ****p* < 0.001. Statistical test: one-way ANOVA with Newman–Keuls post-hoc test. A: *N* = 5; B, D: *N* = 3; C: *N* = 6.

**Figure 8 ijms-21-03963-f008:**
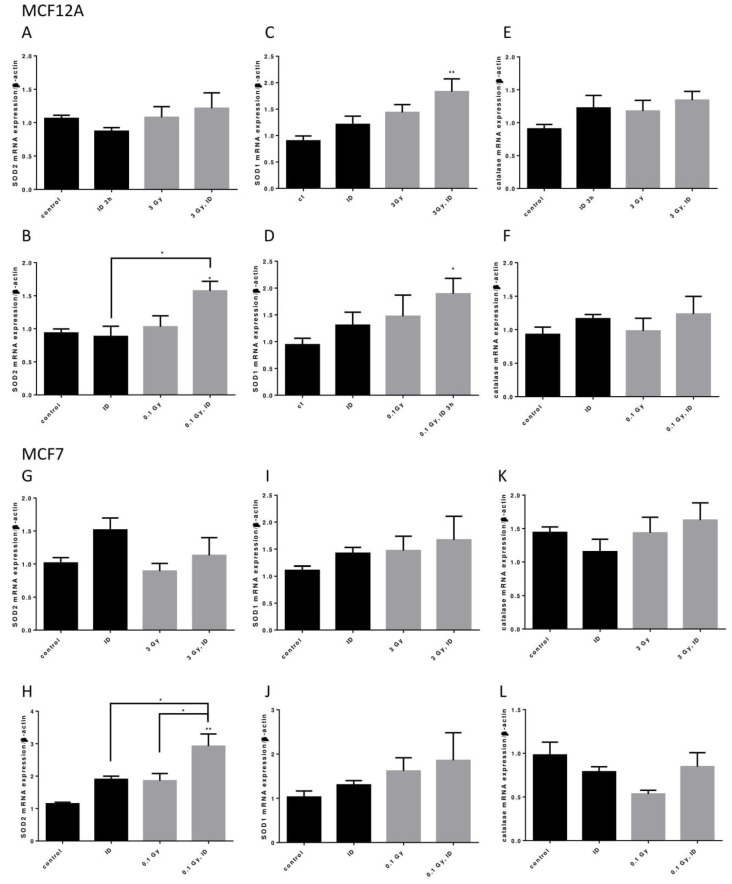
ID and radiation additively up-regulate antioxidant enzymes mRNA in breast cells. Cells were cultured in iodine-containing medium for 7 days. Medium was replaced by iodine-containing (control) or iodine deficient (ID) medium and cells were then irradiated with a dose of 0.1 or 3 Gy. Cells were harvested 3 (MCF12A) or 6 (MCF7) hours after medium change. SOD1, SOD2, and catalase mRNA expression in MCF12A (**A**–**F**) and MCF7 cells (**G**–**L**) was determined using RT-qPCR. Data are expressed as means ± SEM. *p*-value < 0.05 were considered as statistically significant. **p* < 0.05, ***p* < 0.01. Statistical test: one-way ANOVA with Tukey post-hoc test. A: *N* = 6; B, G: *N* = 4; C, D, F, K: *N* = 5; E: *N* = 8; H, I, J, L: *N* = 3.

**Table 1 ijms-21-03963-t001:** Primer sequences and their annealing temperature for qRT-PCR.

RNA	Forward Primer	Reverse Primer	Annealing Temp.
β-actin	5′CATCCTGCGTCTGGACCT3′	5′AGGAGGAGCAATGATCTTGAT3′	62 °C
VEGF-A	5′GCAGATGTCCCGGCGAAGAGAAGA3′	5′CGGGGAGGGCAGAGCTGAGTGTTA3′	62 °C
VEGF165	5′GAGCAAGACAAGAAAATCCC3′	5′CCTCGGCTTGTCACATCTG3′	58.5 °C
VEGF165b	5′GAGCAAGACAAGAAAATCCC3′	5′GTGAGAGATCTGCAAGTACG3′	59 °C
SOD1	5′GCGTGGCCTAGCGAGTTAT3′	5′TTTGCCCAAGTCATCTGCT3′	56 °C
SOD2	5′GTTGGGGTTGGCTTGGTTTC3′	5′ACGATCGTGGTTTACTTTTTG3′	52 °C
Catalase	5′GCAAACCGCACGCTATGG3′	5′ACGAGGGTCCCGAACTGTGT3′	55 °C

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
