# Peer review of "Modulation of VEGF Expression and Oxidative Stress Response by Iodine Deficiency in Irradiated Cancerous and Non-Cancerous Breast Cells"

_ijms, 2020, doi:10.3390/ijms21113963_

Round 1

Reviewer 1 Report

The manuscript by Vanderstraeten describes the effect of iodine deficiency and radiation on VEGF expression and oxidative stress response. The specific concerns are as follows:

  1. Since the results in MCF-7 and MCF12A are different, other cell lines may be considered.
  2. How to interpret the different results observed in MCF-7 and MCF12A?
  3. The authors show that ID induced both ROS and antioxidant enzyme, so what is net result of oxidative stress? How to interpret it?

Reviewer 2 Report

The original manuscript titled “Modulation of VEGF expression and oxidative stress response by iodine deficiency in irradiated cancerous and non-cancerous breast cells” by Jessica Vanderstraeten et al, deeply enlarge a previous work trying to dissect a different mechanism of radiation response hypothetically modulated by iodine status in a cancerous breast cancer cell line (MCF7) vs a normal one (MCF12A). The conclusion is the presence of an addictive effect of acute iodine deprivation and radiation exposure inducing VEGF expression and ROS production.

The reviewer regrets to say the manuscript needs a deep revision and can not be accepted in this version. In principle, the assumption that VEGF expression follows ROS production is not so clearly corroborated by the presented data.

Major points

Figure 1= After any medium change, floating cells should be considered as well as still attached cells. Omitting those cells could potentially and strongly affects every subsequent determination, thus limiting the interests of readers.

Beta-actin is not a preferential house-keeping gene being radiation modulated itself.

Please, add a comment on the hypothetical presence of a sort of threshold of radiation induced VEGF RNA expression between 0,1 and 3 Gy exposure.

Figure 2. = Please, use the same scale for the Y axes in every graph, for the sake of clarity.

Data not in agreement with text: MCF12A cells seem affected by BOTH DOSES of radiation, again with a kind of threshold in term of VEGF expression independently exerted by dose. In MCF7, the differential VEGF expression should be discussed in more details, since it is lower at 3 Gy exposure than control and thus could be additive with acute ID (similarly to MCF12A cells).

Figure 3 = Please, explain the rationale why protein level have been detected 1 hour later than RNA expression. Please, modify the text accordingly (Results section, line 79).

Data not in agreement with text: In the MCF12A the VEGF protein expression does not correlate with RNA. In the lower dose depicted (panel A) the protein is unchanged; in the panel B is increasing but it start from a lower level of VEGF protein in the control itself. Why is there such discrepancies between both controls depicted in panel A, B? Also in the MCF7 panel C there some discrepancies between RNA levels and protein, since the 0,1 Gy lane of protein is comparable to the others, and this is not corroborated by RNA expression.

Figure 5. = The pictures are of poor quality, please try acquiring others with better resolution.

Data not in agreement with text: These figures underline a question mark if VEGF expression does correlate with ROS production. The protein quantification depicted in figure 3 seems not justified by the differences showed by HNE fluorescence (For example, in the MCF12A cells, the level of VEGF in control is almost equal in the WB, but quite different in the HNE fluorescence. The very bright fluorescence of the panel F seems too higher. In the MCF7 cell line, panels K. L looks like of equal fluorescence instead of fig. 3, panel D of WB analysis). Moreover, the hypothetical threshold between the lower dose of 0,1 Gy vs 3 Gy inducing VEGF appears not confirmed by HNE fluorescence, at least in MCF7 cell line (see fig. 5, panels I, K).

From line 158, the manuscript is based on some assumptions that are requested to be further confirmed, and, however, dissecting the role of ROS in breast cancer cell lines is well described in previous literature. As final discrepancy recorded so far, from line 168 to 170 (However, no significant difference in ROS content was observed between iodine-sufficient and iodine-deprived cells when the dye was added before the irradiation (Figure 6b)), I would like to underline the statistical significant difference of p<0,05 indicated in the panel and legend.

Round 2

Reviewer 1 Report

the authors have addressed my concerns.

Reviewer 2 Report

The authors addressed all the raised point.